# Combined Treatment of Heteronemin and Tetrac Induces Antiproliferation in Oral Cancer Cells

**DOI:** 10.3390/md18070348

**Published:** 2020-07-02

**Authors:** Chi-Hung Huang, Tung-Yung Huang, Wong-Jin Chang, Yi-shin Pan, Hung-Ru Chu, Zi-Lin Li, Sukanya Unson, Yu-Tang Chin, Chi-Yu Lin, Haw-Ming Huang, Chao-Nan Hsiung, Fabio Gionfra, Paolo De Vito, Jens Z. Pedersen, Sandra Incerpi, Yi-Ru Chen, Sheng-Yang Lee, Hung-Yun Lin, Paul J. Davis, Jacqueline Whang-Peng, Kuan Wang

**Affiliations:** 1School of Dentistry, College of Oral Medicine, Taipei Medical University, Taipei 11031, Taiwan; hchbox@cgh.org.tw (C.-H.H.); yutangchin@gmail.com (Y.-T.C.); alexlin1018@gmail.com (C.-Y.L.); hhm@tmu.edu.tw (H.-M.H.); 2Division of Cardiology, Department of Internal Medicine, Cathay General Hospital, Taipei 10630, Taiwan; 3Graduate Institute of Cancer Biology and Drug Discovery, College of Medical Science and Technology, Taipei Medical University, Taipei 11031, Taiwan; charvel0203@gmail.com (T.-Y.H.); wjchang@tmu.edu.tw (W.-J.C.); extraganoderma@gmail.com (Y.-s.P.); a0918362166@tmu.edu.tw (H.-R.C.); lizilin0919@gmail.com (Z.-L.L.); sukanya_unson@hotmail.com (S.U.); linhy@tmu.edu.tw (H.-Y.L.); jqwpeng@nhri.org.tw (J.W.-P.); 4Graduate Institute of Nanomedicine and Medical Engineering, College of Medical Engineering, Taipei Medical University, Taipei 11031, Taiwan; aquarlus9132@gmail.com (Y.-R.C.); wangk007@gmail.com (K.W.); 5College of Medical Science and Technology, Taipei Medical University, Taipei 11031, Taiwan; juanita@tmu.edu.tw; 6Department of Sciences, University Roma Tre, 00146 Rome, Italy; fabio.gionfra@uniroma3.it (F.G.); sandra.incerpi@uniroma3.it (S.I.); 7Department of Biology, University of Rome Tor Vergata, 00133 Rome, Italy; Paolo.De.Vito@uniroma2.it (P.D.V.); j.z.pedersen@gmail.com (J.Z.P.); 8Department of Dentistry, Wan-Fang Medical Center, Taipei Medical University, Taipei 11031, Taiwan; 9Pharmaceutical Research Institute, Albany College of Pharmacy and Health Sciences, Albany, NY 12208, USA; pdavis.ordwayst@gmail.com; 10Cancer Center, Wan-Fang Medical Center, Taipei Medical University, Taipei 11696, Taiwan; 11TMU Research Center of Cancer Translational Medicine, Taipei Medical University, Taipei 11031, Taiwan; 12Albany Medical College, Albany, NY 12208, USA

**Keywords:** tetrac, heteronemin, oral cancer, antiproliferation

## Abstract

Background: Heteronemin, a marine sesterterpenoid-type natural product, possesses an antiproliferative effect in cancer cells. In addition, heteronemin has been shown to inhibit *p53* expression. Our laboratory has demonstrated that the thyroid hormone deaminated analogue, tetrac, activates *p53* and induces antiproliferation in colorectal cancer. However, such drug mechanisms are still to be studied in oral cancer cells. Methods: We investigated the antiproliferative effects by Cell Counting Kit-8 and flow cytometry. The signal transduction pathway was measured by Western blotting analyses. Quantitative PCR was used to evaluate gene expression regulated by heteronemin, 3,3’,5,5’-tetraiodothyroacetic acid (tetrac), or their combined treatment in oral cancer cells. Results: Heteronemin inhibited not only expression of proliferative genes and *Homo Sapiens Thrombospondin 1* (*THBS-1*) but also cell proliferation in both OEC-M1 and SCC-25 cells. Remarkably, heteronemin increased *TGF-β1* expression in SCC-25 cells. Tetrac suppressed expression of *THBS-1* but not *p53* expression in both cancer cell lines. Furthermore, the synergistic effect of tetrac and heteronemin inhibited ERK1/2 activation and heteronemin also blocked STAT3 signaling. Combined treatment increased p53 protein and p53 activation accumulation although heteronemin inhibited p53 expression in both cancer cell lines. The combined treatment induced antiproliferation synergistically more than a single agent. Conclusions: Both heteronemin and tetrac inhibited ERK1/2 activation and increased p53 phosphorylation. They also inhibited *THBS-1* expression. Moreover, tetrac suppressed *TGF-β* expression combined with heteronemin to further enhance antiproliferation and anti-metastasis in oral cancer cells.

## 1. Introduction

Oral cancer is a fatal disease, and its incidence is increasing in Taiwan [1]. It accounts for the fourth-highest incidence of malignancy in males and the seventh-highest in the general population of Taiwan [2]. About 95% of oral cancer in Taiwan is oral squamous cell carcinoma (OSCC). Regrettably, about 50% of new OSCC cases presenting to medical centers are stage III or IV cancer lesions. Those patients have a low 5-year survival rate [1]. Current treatments of oral cancer vary depending on the type, location, and stage of cancer at diagnosis. Treatment of the early stages of oral cancer generally is surgery or radiation therapy. Advanced-stage treatment is usually a combination of chemotherapy and radiation therapy which is demonstrated to be more effective for cancer treatment than a single drug. Therefore, searching for novel oral cancer therapies with low side effects might be an important issue in oral cancer therapy.

Heteronemin is a spongean sesterterpenoid, which acts against different kinds of cancers [2,3] with low or non-cytotoxicity to non-malignant cells [4,5]. Heteronemin potently inhibits anchorage-independent growth of human prostate cancer cells [6]. It activates both extrinsic caspase (CASP)-8- and intrinsic caspase-9-dependent pathways in prostate cancer cells to induce apoptosis [6]. Heteronemin directly generates reactive oxygen species (ROS) to induce oxidative stress, which induces phosphorylated talin and promotes antiproliferation in cancer cells [3], but it does not disrupt talin/focal adhesion kinase (FAK) complex formation. In addition, heteronemin can cause morphological changes via interfering with actin microfilaments [3]. Heteronemin is shown to prevent phosphorylation of c-Met/Src/signal transducer and activator of transcription 3 (STAT3) [6]. Furthermore, it suppresses expression of STAT3-regulated genes such as *Bcl-xL*, *Bcl-2*, and *cyclin D1* [2]. It efficiently antagonizes hepatocyte growth factor (HGF)-stimulated c-Met/STAT3 activation, and the proliferation and colony formation of refractory prostate cancer cells [6]. Our results indicated that heteronemin concurrently inhibits *TGF-β* expression with antiproliferation, anti-migration, and anti-adhesion effects [2]. On the other hand, heteronemin inhibits *p53* expression and activity in cholangiocarcinomas [2].

The thyroid hormone deaminated analog, 3,3’,5,5’-tetraiodothyroacetic acid (tetrac), inhibits cancer cell growth in vitro and in animal xenografts [7,8] and also has been shown to have no cytotoxicity to non-malignant cells [9]. It induces antiproliferation as well as anti-angiogenesis and anti-metastasis [7,8] via activating expression of pro-apoptotic genes such as *p53*, *p21*, and *CBY1*. It also suppresses the expression of proliferative genes. It has shown to activate *Homo Sapiens Thrombospondin 1* (*THBS-1*) which is anti-angiogenic in MDA-MB-231 and colorectal cancer HCT-116 cells. Cross-talk between different signal transduction pathways plays a critical role in cancer development and therapeutic resistance. Tetrac and nano-diamino-tetrac (NDAT) were shown to inhibit *K-RAS*-mutant cancer cell growth in colorectal cancer and lung cancer [9,10,11]. They are able to compensate for a lack of therapeutic efficacy in gefitinib- and cetuximab-resistant colorectal cancer cells [9,11,12].

Against this background, we investigated mechanisms of how tetrac potentiates anti-cancer growth induced by heteronemin in oral cancer cells. Results reported here indicate that heteronemin induces antiproliferation in oral cancer cells by inhibiting activation of extracellular signal-regulated kinase 1/2 (ERK1/2) and STAT3. Heteronemin suppressed expression of proliferative genes, *THBS-1* and *p53*, but not p53 protein accumulation. It also suppressed *TGF-β* expression in OEC-M1 cells but not in SCC-25 cells. On the other hand, tetrac enhanced heteronemin-induced antiproliferation via inhibiting ERK1/2 activation. It further inhibited expression of *THBS-1* in both cancer cell lines. Tetrac suppressed *TGF*-β expression in combination with heteronemin. It promoted p53 phosphorylation in combination with heteronemin. The mechanisms implicated in the combined treatment of tetrac and heteronemin showed a synergistic antiproliferative effect.

## 2. Results

### 2.1. Heteronemin Induces Antiproliferation and Modulates Gene Expression in Different Types of Oral Cancer Cells

Heteronemin has been shown to induce antiproliferation in human cholangiocarcinoma cells and various cancer cells [2]. It has been shown to suppress expression of *TGF-β* and *p53* in cholangiocarcinoma [2]. Studies were conducted to examine the growth inhibition of heteronemin in two different types of oral cancer cells. OEC-M1 or SCC-25 cells were treated with different concentrations of heteronemin for 24, 48, and 72 h. Then the Cell Counting Kit-8 reagent was added to detect the cytoxicity after treatment (Figure 1). In the time-course experiment, heteronemin caused a significant cytotoxic effect in both oral cancer cell lines, starting at 0.313 μM, in a dose-dependent manner (Figure 1A,B). Furthermore, prolonged treatment increased the cytotoxic effect of herteronemin. The inhibitory rate for each concentration is shown in Figures in the appendix. To further understand the mechanisms involved in heteronemin-induced antiproliferation, SCC-25 cells were treated with different concentrations of heteronemin for 24 h and harvested for a flow cytometric assay (Figure 1C). Low concentration of heteronemin treatment (0.313 μM) mildly increased the cell population in G0/G1 phase and decreased the cell population in S and G2/M phase. With increasing concentration of heteronemin, there was cell phase arrest in the G2/M phase (0.625 μM) and drastically increased sub-G0/G1 population at the highest concentration of heteronemin (1.25 μM). These results suggest that heteronemin may activate different pathways to induce antiproliferation at different concentrations.

To investigate the potential mechanism of heteronemin-induced antiproliferation in oral cancer cells, we further studied the effect of heteronemin on gene expression in OEC-M1 and SCC-25 cells. Overall, heteronemin suppressed expression of *p53*, *PCNA*, and *THSB-1* starting significantly at 0.313 µM in a concentration-dependent manner, except for *CCND1* in OEC-M1 cells (Figure 2). The pro-apoptotic *p21* expression was enhanced by heteronemin in a dose-dependent manner. Following the expression of *PCNA* and *CCND1*, these changes suggest that the cell cycle of OEC-M1 is indeed inhibited by heteronemin. *TGF-β1* expression was only inhibited by heteronemin at 1.25 µM.

Moreover, the same genes were examined in another oral cancer cell line, SCC-25 cells. The qPCR results indicated that heteronemin attenuated expression of the proliferative gene, *PCNA*, and *CCND1*, at 0.313 µM, but it was not much more effective as the concentration increased. This situation was also observed in *p53* and *THBS-1*. The expression of *p21* was increased dose-dependently by heteronemin in SCC-25 as well as OEC-M1. Unlike OEC-M1, heteronemin stimulated high expression of *TGF-β1* starting at 0.625 µM.

Although heteronemin induced antiproliferation in two oral cancer cell lines, it suppressed *p53* expression in both cancer cell lines and induced *TGF-β* expression in SCC-25 cells. Tetrac has been shown to activate *p53* expression [10] and inhibit *TGF-β* expression [13]. Additionally, thyroid hormone stimulates *TGF-β* expression in oral cancer cells [14] which can be inhibited by tetrac. Thus, the combined treatment of tetrac and heteronemin might compensate for heteronemin-induced antiproliferative activities via suppressing *TGF-β* and enhancing *p53* expression in oral cancer cells. Next, we investigated the possible mechanisms involved in the combined treatments.

### 2.2. Tetrac and Heteronemin Inhibit Signal Transduction Pathways in Oral Cancer Cells

We previously showed that inhibition of STAT3 and ERK1/2 activation plays an important role in antiproliferation in cancer cells [7,10,15]. We investigated the mechanisms involved in the heteronemin-induced anti-cancer ability in oral cells. To investigate the signal transduction pathways involved in antiproliferation induced by the combination of tetrac and heteronemin, OEC-M1 and SCC-25 cells were treated with 10^−7^ M tetrac and different concentration of heteronemin, and their combination for 24 h. Then cells were extracted and quantified by RIPA buffer and BCA Protein Assay Kit (Pierce, Rockford, IL, USA), respectively. Western blotting analysis was used to detect the phosphorylation and total protein of STAT3 and ERK1/2 (Figure 3). Tetrac did not affect status of total STAT3 or phosphorylated STAT3 in both cancer cell lines (Figure 3). Heteronemin suppressed phosphorylation of STAT3 activation in OEC-M1 but increased total form starting at 0.625 µM (Figure 3 3A, upper panel). The combined treatment showed the highest total STAT3 at 0.313 µM. Combination treatment with a higher dosage of heteronemin did not further inhibit STAT3 activation. Tetrac (10^−7^ M) inhibited ERK1/2 activation significantly in both oral cancer cell lines (Figure 3A,B). Heteronemin inhibited phosphorylation of ERK1/2 activation in a concentration-dependent manner in OEC-M1 cells (Figure 3A, lower panel). The combined treatment at 0.313 µM heteronemin with tetrac showed significant decrease as compared with single-agent alone. Parallel studies conducted in SCC-25 cells showed comparable results. Unlike OEC-M1, both phosphorylated STAT3 and total protein of STAT3 were decreased in a dose-dependent manner by heteronemin treatment in SCC-25 (Figure 3B, upper panel). The combined treatment also revealed a decrease in the accumulation of total STAT3. Heteronemin alone significantly reduced phosphorylated ERK1/2, and especially, completely suppressed ERK1/2 activation at 1.25 µM (Figure 3B, lower panel). Furthermore, heteronemin combined with tetrac showed the same trend of inhibiting ERK1/2 activation. However, there was no synergistic effect since the inhibitory effect was maximal. These results indicated that both heteronemin and tetrac inhibited ERK1/2 activation and heteronemin further blocked STAT3 phosphorylation. The inhibitory effects by heteronemin and tetrac may play important roles in antiproliferation in oral cancer cells.

We further investigated the effects of combined treatment of 10^−7^ M tetrac and 0.313 µM heteronemin on expression of genes involved in cell cycle, proliferation, pro-apoptosis, and angiogenesis. Simultaneous treatment with both heteronemin and tetrac suppressed *PCNA* and *CCND1* expression more significantly than single treatment with tetrac or heteronemin in both OEC-M1 and SCC-25 cells. The expression of *p21* was also increased by combination treatment of tetrac and heteronemin, however, the combination treatment did not show further effect on *p21* gene expression than heteronemin single treatment (Figure 4).

Heteronemin did not affect *TGF-β1* expression significantly, but tetrac treatment and the combination treatment inhibited the expression of *TGF-β1* in OEC-M1 cells (Figure 5). Both tetrac treatment and the combination treatment inhibited the expression of genes involved in angiogenesis and metastasis, *THBS-1* and *MMP9*. On the other hand, heteronemin alone did not affect expression of *THBS-1* and *MMP9* in SCC-25 cells. Heteronemin increased expression of *TGF-β1* in SCC-25 cells but this inductive effect of heteronemin was compensated for in the combination with tetrac treatment (Figure 5).

### 2.3. Tetrac and Heteronemin Synergistically Increase p53 Accumulation and Induce Antiproliferation in Oral Cancer Cells

Since heteronemin inhibited *p53* expression in cholangiocarcinoma [2] and in both oral cancer cell lines (Figure 2), we examined if the combined treatment of 10^−7^ M tetrac with different concentrations of heteronemin would change *p53* expression. Tetrac increased *p53* expression slightly in OEC-M1 cells but significantly in SCC-25 cells (Figure 6A,B, upper panel). However, tetrac did not increase *p53* expression in the combination with heteronemin. On the other hand, total p53 protein increased in 0.625 μM heteronemin-treated OEC-M1 cells and the combination increased p53 accumulation at concentrations of 0.313 and 0.625 μM of heteronemin (Figure 6A, lower panel). Surprisingly, in SCC-25 cells, heteronemin treatment increased p53 accumulation and phosphorylation. Tetrac enhanced p53 accumulation and phosphorylation maximally at 0.625 µM heteronemin (Figure 6B, lower panel).

We examined the antiproliferative effect of combined treatment of tetrac and heteronemin. OEC-M1 or SCC-25 cells were treated with different concentrations of heteronemin in the presence or absence of tetrac (10^−7^ M) for 48 h. Although tetrac did not inhibit the cell growth of OEC-M1, the combined treatment with heteronemin (0.313 µM and 0.625 µM) and tetrac had a synergetic effect on antiproliferation in OEC-M1 cells (Figure 7A). Similar results were observed in SCC-25 cells (Figure 7B).

To understand whether tetrac or heteronemin has a direct functional role in the regulation of cell cycle and apoptosis in oral cancer cells, we performed propidium iodide (PI) staining for further exploration. The percentage of each cell phase stage in OEC-M1 and SCC-25 are shown in Figure 8. The percentages of cell stages in OEC-M1 and SCC-25 cells showed a different trend. The percentage G0/G1 in OEC-M1 was increased with heteronemin but did not increase in the combination group (Figure 8A). However, the sub-G0/G1 phase of OEC-M1 cells in the combined treatment was higher than with heteronemin alone (0.625 μM). Thus the results indicated that the levels of apoptotic and dead cells were increased by heteronemin combined with tetrac. Then again, the acute cell death of OEC-M1 induced by high concentration of heteronemin might have been due to drug susceptibility (data not shown). The cell phase of SCC-25 tended to G2/M after heteronemin treatment (Figure 8B). Moreover, a similar phenomenon was observed in the combined heteronemin and tetrac treatment. Interestingly, there was a ten percent raise in sub-G0/G1 phase in 1.25 μM heteronemin-treated SCC-25 cells compared with the untreated control, but 2% of sub-G0/G1 was rescued when tetrac was present (data not shown). Therefore, in the heteronemin and combined treatment groups, OEC-M1 and SCC-25 showed G0/G1 and G2/M arrest, respectively. Additionally, both of them increased the value of sub-G0/G1 phase at 0.313 and 0.625 μM conditions. Our results revealed the drug-dependent induction of apoptosis in the combination treatment.

## 3. Discussion

A deaminated analog of thyroxine (T_4_), tetrac, and NDAT can inhibit cancer proliferation via the receptor of the cell surface integrin, αvβ3. Tetrac and NDAT reduce expression of pro-inflammatory genes, such as *interleukin* (*IL*)-*1α*, *IL-1β*, and *IL-6* [16]. They increase expression of *IL-11*, a desirable stimulator of hematopoietic stem cell proliferation [16]. Additionally, NDAT downregulates expression of the apoptosis inhibitors, *X-linked inhibitor of apoptosis (XIAP)* and *myeloid cell leukemia sequence 1 (MCL1)* and upregulates apoptosis-promoting *CASP2* and *BCL2L14*. Tetrac and NDAT increase *THBS-1* and *CBY1*, a nuclear inhibitor of β-catenin activity in various types of cancer cells [17,18]. Both tetrac and NDAT were shown to downregulate expression of *K-Ras* and *EGFR*. Additionally, they potentiate other anticancer agents such as cetuximab- [11] or gefitinib-induced [12] anticancer activities. They are also used to compensate natural products such as resveratrol-induced anti-cancer treatment in colorectal cancers in in vitro and in vivo xenograft models [3,7]. Additionally, NDAT was shown to enhance resveratrol-induced antiproliferation in oral cancer cells [15]. In current studies, tetrac was able to inhibit expression of *CCND1*, *PCNA*, and *THBS-1* in both OEC-M1 and SCC-25 cells (Figure 4). *THBS-1* has been shown to play a tumorigenic role in oral cancers [19].

Tetrac inhibits ERK1/2 activation to inhibit cancer proliferation that has been well-established [10]. Heteronemin is an inhibitor of STAT3 [6]. Heteronemin inhibited not only STAT3 but also ERK1/2 activation in both OEC-M1 cells and SCC-25 cells (Figure 3). Tetrac inhibited ERK1/2 but not STAT3 activation in OEC-M1 cells and SCC-25 cells (Figure 3). These results suggest that heteronemin and tetrac may induce antiproliferation via a different signal transduction pathway. This phenomenon was observed in our previous studies. Resveratrol induces antiproliferation via ERK1/2 activation [20] while tetrac inhibits cell proliferation via ERK1/2 inhibition. Tetrac potentiates resveratrol-induced antiproliferation without affecting resveratrol-induced ERK1/2 activation [3]. Studies conducted by Ho Y., et al. also indicate that resveratrol induced ERK1/2 activation and NDAT inhibited ERK1/2 activation, but their combination induced antiproliferation of oral cancer cells [15].

Heteronemin inhibited proliferation (Figure 1) and proliferative gene expression (Figure 2) in both oral cancer cells. It also inhibited expression of *PCNA*, *CCND1*, and *THBS-1*. Pro-apoptotic *p21* expression was enhanced by heteronemin in a concentration-dependent manner. Interestingly, heteronemin at 0.313 µM significantly suppressed expression of *THBS-1* and *p53* (Figure 2). Heteronemin and tetrac both inhibited *THBS-1* expression (Figure 2, Figure 4 and Figure 5). Tetrac was shown to stimulate *THBS-1* expression in other types of cancer cells such as breast cancer [21], medullary carcinoma of the thyroid [22], and pancreatic cancer [23]. However, expression of *THBS-1* was demonstrated to be induced by *TGF-β1* in cancer stroma and to promote invasion of oral squamous cell carcinoma [19]. The combined treatment of tetrac and heteronemin more strongly reduced *TGF-β1* expression (Figure 5).

Both tetrac and heteronemin inhibited *TGF-β1* expression (Figure 5). Knocking-down *TGF-β1* enhanced heteronemin-induced antiproliferation [24]. These results indicate that co-treatment with tetrac and heteronemin induced antiproliferation via *TGF-β* suppression. TGF-β acts as a tumor promoter in the progressive stages of cancers to support tumor cell proliferation, invasion, metastasis, and immune evasion [25,26,27,28,29]. TGF-β forms a heterotetrameric complex with TGF-βR1 thereby initiating binding with the TGF-β receptor II (TGF-βRII). Successively, TGF-βR1 is phosphorylated by TGF-βRII to recruit and phosphorylate the cytosolic transcription factors, SMAD2 and SMAD3 [30,31,32]. Phosphorylated SMAD2/3 forms a complex with co-SMAD (SMAD4) after dissociation from TGF-βRs. The formed transcriptional complex is translocated into the nucleus and regulates expression of numerous target genes [33,34].

Additionally, TGF-β signaling also activates several non-SMAD pathways, for example PI3K/AKT, INK/p38, Ras-ERK, and RhoA pathways [25,34]. Heteronemin inhibited both pERK1/2 activation and STAT3 phosphorylation (Figure 3). Moreover, heteronemin promoted *TGF-β1* expression in OEC-M1 cells (Figure 5) which may have also reduced TGF-β1-induced activation of ERK1/2 and STAT3. Although heteronemin did not inhibit TGF-β1 expression in SCC-25 cells, tetrac also inhibited *TGF-β1* expression which may compensate the *TGF-β1* expression effect of heteronemin in SCC-25 cells.

Interestingly, TGF-β was shown to possess antitumor activity [35]. The crosstalk between p53 signaling and TGF-β is well documented [36]. The antitumor effect of TGF-β signaling stabilizes the accumulation and recruitment of wild-type p53 to its DNA-binding sites. Inactivation of p53 interrupts TGF-β-induced cellular activities. However, crosstalk between p53 and TGF-β signaling demonstrates that p53 can act as a component of SMAD complexes that participates in the stabilization of SMAD–DNA complexes and modulates various tumor suppressor genes [37,38,39,40,41]. In addition, heteronemin reduced *p53* expression (Figure 2). Knockdown of *p53* did not reduce or affect the abundance of *TGF-β* [2], indicating that p53 might not be associated with TGF-β forming a complex with, and stabilizing p53 in oral cancers. Heteronemin reduced expression of both *TGF-β1* and *p53* in oral cancer cells (Figure 2) and cholangiocarcinomas [2]. Together, the results suggest that heteronemin suppressed oncogenic TGF-β1 expression to induce antiproliferation in which p53 does not play a role in oral cancer cells.

Tetrac did not affect *p53* expression, but heteronemin inhibited *p53* expression in OEC-M1 cells and SCC-25 cells (Figure 6). Combined treatment with tetrac and heteronemin even downregulated p53 expression (Figure 6). However, tetrac, 0.625 µM heteronemin, or the combination increased total p53 amount in OEC-M1 cells. Tetrac, heteronemin, or the combination increased not only total p53 amount but also phosphorylated p53 in SCC-25 cells. These results suggest that combined tetrac and heteronemin may stabilize p53 or increase its phosphorylation.

There were different trends of cell cycles between OE-CM1 and SSC-25 treated with heteronemin and tetrac (Figure 8). These two drugs had been shown to inhibit cancer cell growth in the previous studies. Our results further confirmed the population of sub-G0/G1 in oral cancer cells was increased by tetrac or heteronemin treatment alone. Heteronemin and tetrac showed a significantly synergistic growth-inhibitory effect on OEC-M1 and SCC-25 cells under the combined treatment group of 0.625 μM heteronemin. The phenomena of cell cycle are consistent with our experiments in cell proliferation (Figure 7). Although high concentration of heteronemin (1.25 μM) caused more cell death in these cells, the sub-G0/G1 and G0/G1 of SCC-25 could be rescued by tetrac. For heteronemin or combined treatment, increasing cell cycle arrest of G0/G1 and G2/M was observed in OEC-M1 and SCC-25, respectively. Thus OEC-M1 and SCC-25 showed different trends in populations of cell cycle. We suggested that the difference is caused by a resistant gene or p53 mutation in SCC-25 cells after drug treatment.

Compared with the cell cycle of OEC-M1, it suggested that the p53 mutation of SCC-25 may be one of the reasons for the different trends between them. Based on RT-PCR data and Western blot analysis, the differential expression of p21, p53, and CCND1 in oral cancer cells after heteronemin or tetrac treatment may play critical roles in the G0/G1 cell cycle arrest that blocks cell proliferation and induces apoptosis.

## 4. Materials and Methods

### 4.1. Cell Culture

An indigenous human oral epidermoid carcinoma OEC-M1 cell was established from a gingival epidermal carcinoma of a patient in Taiwan by Dr. Ching-Liang Meng [42]. This cell line was a gift from Dr. Hsien-Chung Chiu. Human squamous carcinoma of the tongue SCC-25 cells (ATCC^®^ CRL-1628™) were obtained from the American Type Culture Collection (ATCC; Manassas, VA, USA). Cells were tested and authenticated with Bioresource Collection and Research Center of Taiwan (through an isoenzyme analysis, mycoplasma, cytogenetics, tumorigenesis, and receptor expression testing). OEC-M1 and SCC-25 cells were respectively maintained in RPMI-1640 and Dulbecco’s modified Eagle medium (DMEM)/F12 supplemented with 10% fetal bovine serum (FBS) at 37 ℃ in a humidified incubator with 5% CO_2_ and 95% air. Cells at no later than passage 15 were used for the experiments. Before the treatment, cells were placed in serum-free medium for 24 h. In treatment condition, cells were incubated in respective medium with 2% FBS.

### 4.2. Flow Cytometric Analysis

Oral cancer cells were grown in 100-mm tissue culture dishes until 80% confluent and treated daily with different reagents for 24 h with refreshed medium containing reagents. Cells were harvested by trypsinization, washed with phosphate-buffered saline (PBS) and re-suspended in 200 μL PBS (10^5^ to 10^6^ cells). To quantify cellular DNA contents, cells were permeabilized by fixation with 70% ethanol for 30 min at 4 ℃. Samples were stored in 70% ethanol at −20 ℃ for up to two weeks prior to propidium iodide (PI) staining and the flow cytometric analysis. 1 mL 0.5% triton buffer containing RNase A (0.05%) was added to the cell suspension, and incubation was carried out at 37 ℃ for 1 h; cells were then stained PI (50 μg/mL) and maintained in the dark at room temperature for 30 min. Flow cytometry was carried out on an Invitrogen Attune™ NxT Acoustic Focusing Cytometer (ThermoFisher Scientific, Waltham, MA, USA) instrument. The percentages of DNA content were analyzed using Attune NxT Flow Cytometer software (version 4.2) to determine the fractions in each phase of cell cycle (G0/G1, S, and G2/M).

### 4.3. Cell Proliferation Assay

Cell proliferation was evaluated using the Cell Counting Kit-8 (cat. no.: 96992, Sigma-Aldrich, St. Louis, MO, USA). The assay was performed in 96-well microplates with phenol red medium (RPMI-1640: OEC-M1; DMEM/F12: SCC-25). Cells were seeded into 96-well plates at a density of 1.5 × 10^3^ cells/well and cultured in 5% CO_2_ at 37 ℃ overnight. On the second day, the medium was replaced with fresh medium without FBS, and then incubated for 24 h at 37 ℃. ‘Various concentrations of tetrac and heteronemin were added, and six wells were arranged in one column for each condition. Media with or without the drug were refreshed every 2 days. For the detection step, the CCK-8 reagent (10 µL) was added to each well after 2 days of drug treatment and incubated for 1 h at 37 ℃. The optical absorbance was read with a VersaMax Microplate reader (Molecular Device, San Jose, CA, USA) at a wavelength of 450 nm. The readings were normalized to blank medium. Cell proliferation was calculated as the ratio of the readings of the treatment group to the control group.

### 4.4. Real-Time Quantitative Polymerase Chain Reaction (qPCR)

Total RNA was extracted and genomic DNA was removed with an Illustra RNAspin Mini RNA Isolation Kit (GE Healthcare Life Sciences, Buckinghamshire, UK). One microgram of DNase I-treated total RNA was reverse-transcribed with a RevertAid H Minus First Strand cDNA Synthesis Kit (Thermo Fisher Scientific) into complementary (c)DNA and used as the template for the real-time PCRs and analysis. The real-time PCRs were performed using a QuantiNovaTM SYBR^®^ Green PCR Kit (Qiagen, Hilden, Germany) on a CFX Connect™ Real-Time PCR Detection System (Bio-Rad Laboratories, Hercules, CA, USA). This involved initial denaturation at 95 °C for 5 min, followed by 40 cycles of denaturing at 95 °C for 5s and combined annealing/extension at 60 °C for 10s, as detailed in the manufacturer’s instructions. The primer sequences were shown as follows: *Homo sapiens cyclin D1 (CCND1)*, forward 5’-CAAGGCCTGAACCTGAGGAG-3’ and reverse 5’-GATCACTCTGGAGAGGAAGCG-3’ (Accession no.: NM_053056); *Homo sapiens proliferating cell nuclear antigen (PCNA)*, forward 5′-TCTGAGGGCTTCGACACCTA-3′ and reverse 5′-TCATTGCCGGCGCATTTTAG-3′ (Accession No.: BC062439.1); *Homo sapiens* cyclin-dependent kinase inhibitor 1A (p21), forward 5’-CTGGGGATGTCCGTCAGAAC-3’ and reverse 5’-CATTAGCGCATCACAGTCGC-3’ (accession no.: BT006719.1); *Homo sapiens tumor protein p53 (p53)*, forward 5’-AAGTCTAGAGCCACCGTCCA-3’ and reverse 5’-CAGTCTGGCTGCCAATCCA-3’ (Accession No.: NM_000546.5); *Homo sapiens transforming growth factor* (*TGF-β*), forward 5’-GCCCTGGACACCAACTATTGC-3’ and reverse 5’-GCTGCACTTGCAGGAGCGCAC-3’ (accession no.: NM_000660.6); *Homo sapiens thrombospondin 1 (THBS-1)* forward 5’- ATCCTGGACTCGCTGTAGGT-3’ and reverse 5’- GTCATCGTCCCTTTCGGTGT-3’ (Accession no.: NM_003246.6); *Homo sapiens matrix metallopeptidase 9 (MMP-9)*, forward 5’ TGTACCGCTATGGTTACACTCG 3’ and reverse 5’ GGCAGGGACAGTTGCTTCT 3’ (Accession no.: NM_004994.3); *Homo sapiens Catenin beta-1 (β-catenin)* forward 5’- CTGGTCCTTTTTGGTCGAGGA-3’ and reverse 5’- GCAAGGCTAGGGTTTGATAAAT-3’ (Accession no.: NM_001904.4); and *Homo sapiens Caspase 2 (CASP2)* forward 5’-GCATGTACTCCCACCGTTGA-3’ and reverse 5’-GACAGGCGGAGCTTCTTGTA-3’ (Accession no.: NM_032982.4). Calculations of the relative gene expression (normalized to *18S* reference gene) were performed according to the ΔΔCT method. Fidelity of the PCR was determined with a melting temperature analysis.

### 4.5. Western Blotting

Western blot analyses were conducted as previously described [43,44]. In brief, after treatment with tetrac, heteronemin, or their combination, cells were harvested. Total proteins were extracted and quantified. Protein samples were resolved by 10% sodium dodecylsulfate-polyacrylamide gel electrophoresis (SDS-PAGE). A 30-μg quantity of protein was loaded in each well with 5× sample buffer, and samples were resolved with electrophoresis at 100 V for 2 h. The resolved proteins were transferred from the polyacrylamide gel to Millipore Immobilon-PSQ Transfer polyvinylidene difluoride (PVDF) membranes (Millipore, Billerica, MA, USA) with Mini Trans-Blot^®^ Cell (Bio-Rad Laboratories). Membranes were blocked with a solution of 5% skim milk in Tris-buffered saline. Membranes were incubated with primary antibodies to phosphorylated (phospho)-ERK, phospho-STAT3, and their corresponding proteins (Cell Signaling Technology, Beverly, MA, USA), and GAPDH (GeneTex International, Hsinchu City, Taiwan) overnight at 4 °C. Proteins were detected with horseradish peroxidase (HRP)-conjugated secondary antibodies and Immobilon™ Western HRP Substrate Luminol Reagent (Millipore, Burlington, MA USA). Western blots were visualized and recorded with the Amersham Imager 600 system (GE Healthcare Life Sciences, Pittsburgh, PA, USA). The densitometric analysis of Western blots was conducted with Image J 1.47 software (National Institute of Health, Bethesda, MD, USA) according to the software’s instructions.

### 4.6. Quantification of Results and Statistical Analysis

Densities of Western blots and gene expressions of the real-time qPCR were analyzed by IBM SPSS Statistics software version 19.0 (SPSS, Chicago, IL, USA). Student’s t-test was conducted, and changes were considered significant at *p* < 0.05 (*, #, $), 0.01 (**, ##, $$) and 0.001 (***, ###, $$$).

## 5. Conclusions

In conclusion, heteronemin induced antiproliferation in oral cancer cells by inhibiting activation of ERK1/2 and STAT3. Heteronemin suppressed expression of *p53* and *THBS-1*. In addition, heteronemin inhibited *TGF-β*1 expression in OEC-M1 cells but activated *TGF-β1* expression in SCC-25 cells. On the other hand, tetrac did not suppress *p53* expression but inhibited *THBS-1* expression in both oral cancer cell lines. Additionally, tetrac suppressed *TGF-β* expression in combination with heteronemin which may have compensated for heteronemin-induced suppression of *p53*. The different trend of cell population in OEC-M1 and SCC-25 may have been due to p53 mutation, whereas the sub-G0/G1 in two cells was increased after combined treatment. Therefore, combined treatment with heteronemin and tetrac is able to enhance heteronemin-induced antiproliferation in oral cancer cells.

## Figures and Tables

**Figure 1 marinedrugs-18-00348-f001:**
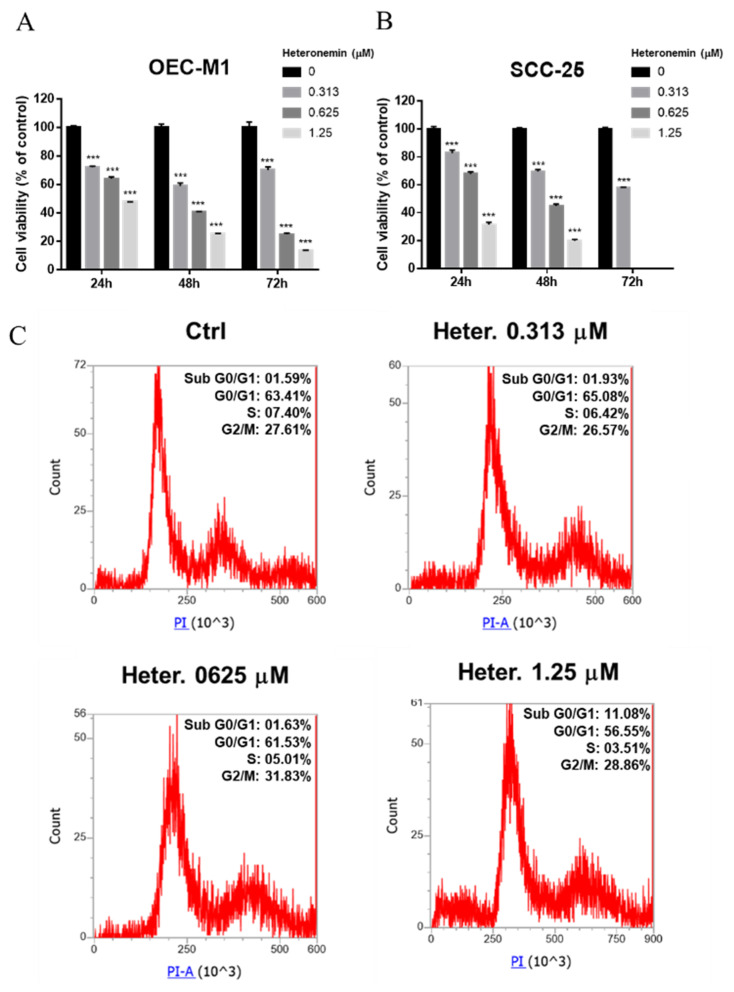
Heteronemin induces antiproliferation and cell cycle analysis in oral cancer cells. OEC-M1 (**A**) and SCC-25 (**B**) cells were treated with different concentrations of heteronemin for 24, 48, and 72 h. Cell proliferation was detected with the Cell Counting Kit-8. Number of independent studies (*n*) = 3. (**C**) SCC-25 cells were treated with different concentrations of heteronemin (Heter.) for 24 h. Cell cycle analysis was conducted as described. Cells were harvested, and flow cytometric assay was conducted as described in the Materials and Methods section. *n* = 3. Data are expressed as mean ± SD. *** *p* < 0.05 compared with untreated control.

**Figure 2 marinedrugs-18-00348-f002:**
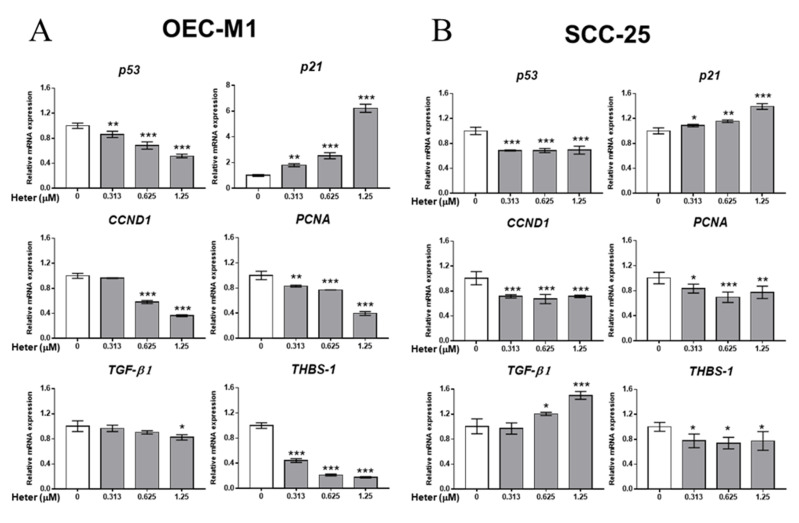
Heteronemin regulates gene expression in oral cancer cells. RNA was extracted from OEC-M1 (**A**) and SCC-25 (**B**) cells at the end of treatment for qPCR analyses of *p53*, *p21*, *CCND1*, *PCNA*, *TGF-β1*, and *THBS-1*. Number of independent studies (*n*) = 3. * *p* < 0.05, ** *p* < 0.01, *** *p* < 0.001, compared to the control for each gene.

**Figure 3 marinedrugs-18-00348-f003:**
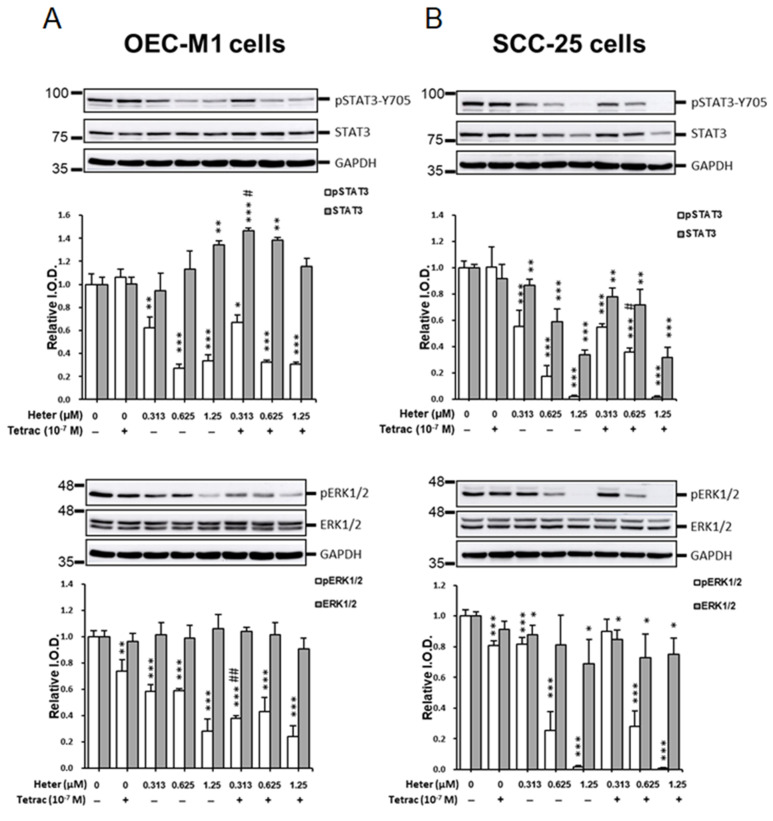
Tetrac and heteronemin inhibit signal transduction pathways in oral cancer cells. OEC-M1 (**A**) and SCC-25 (**B**) cells were treated with tetrac, heteronemin, or their combination for 24 h. Cells were harvested, and total proteins were extracted. Western blot analyses were conducted for pSTAT3, STAT3, pERK1/2, and ERK1/2. GAPDH was used as internal control. Number of independent studies (*n*) = 4. * *p* < 0.05, ** *p* < 0.01, *** *p* < 0.001, compared to the control; ^#^
*p* < 0.05, ^##^
*p* < 0.01 compared to the same dosage of heteronemin treatment only.

**Figure 4 marinedrugs-18-00348-f004:**
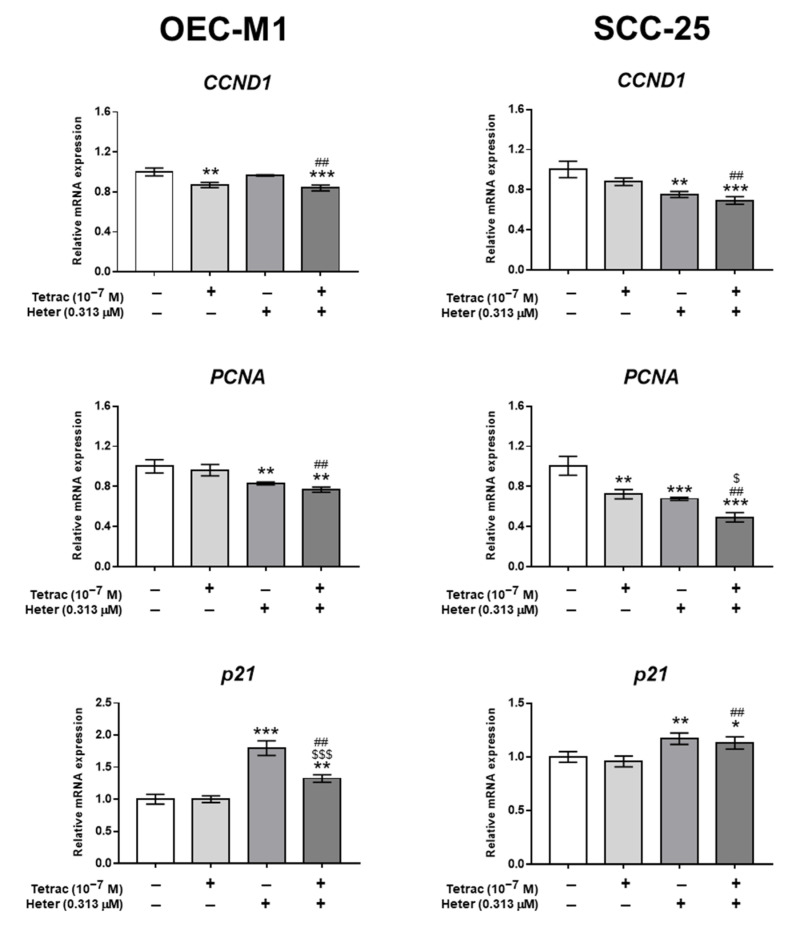
Tetrac and heteronemin regulate cell-cycle-related gene expression in oral cancer cells. OEC-M1 and SCC-25 cells were treated with 0.313 µM heteronemin, 10^−7^ M tetrac, or their combination for 24 h. RNA was extracted and qPCR analyses were conducted for *CCND1*, *PCNA*, and *p21*. Numbers of independent studies (*n*) = 3. * *p* < 0.05, ** *p* < 0.01, *** *p* < 0.001 compared to the control; ^##^
*p* < 0.01 compared to the tetrac treatment only; ^$^
*p* < 0.05, ^$$$^
*p* < 0.001 compared to the same dosage of heteronemin treatment only.

**Figure 5 marinedrugs-18-00348-f005:**
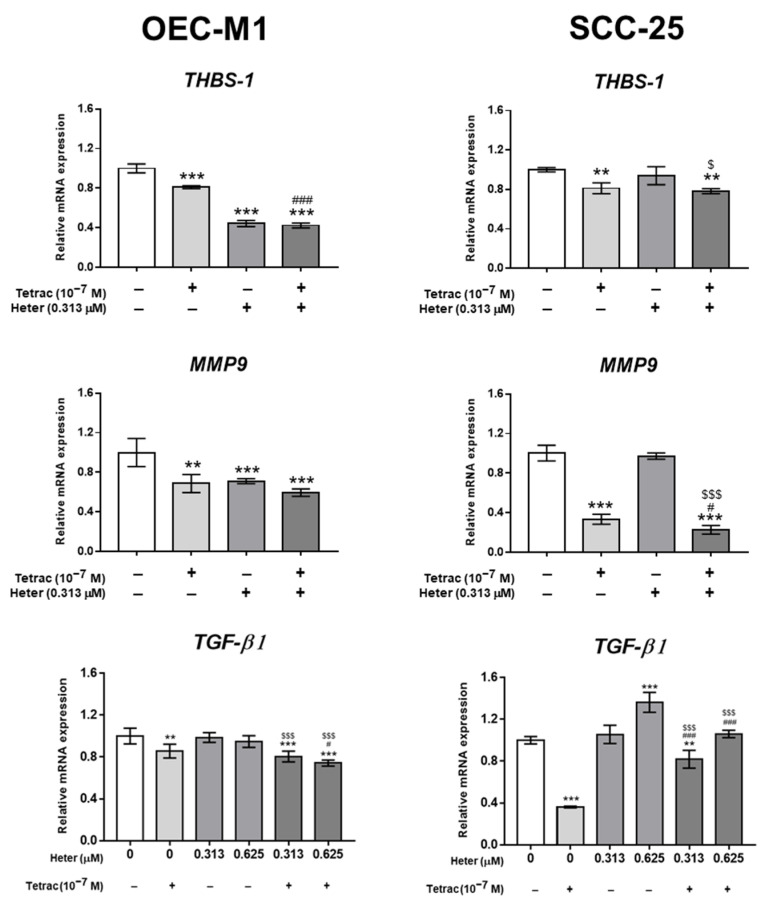
Tetrac and heteronemin regulate gene expression on modulating tumor–stromal function in oral cancer cells. OEC-M1 and SCC-25 cells were treated with 0.313 and 0.625 µM heteronemin, 10^−7^ M tetrac, or their combination for 24 h. RNA was extracted from harvested cells at the end of treatment for qPCR studies of *THBS-1*, *MMP9*, and *TGF-β1*. Numbers of independent studies (*n*) = 3. ** *p* < 0.01, *** *p* < 0.001 compared to the control; ^#^
*p* < 0.05, ^###^
*p* < 0.001 compared to the tetrac treatment only; ^$^
*p* < 0.05, ^$$$^
*p* < 0.001 compared to the same dosage of heteronemin treatment only.

**Figure 6 marinedrugs-18-00348-f006:**
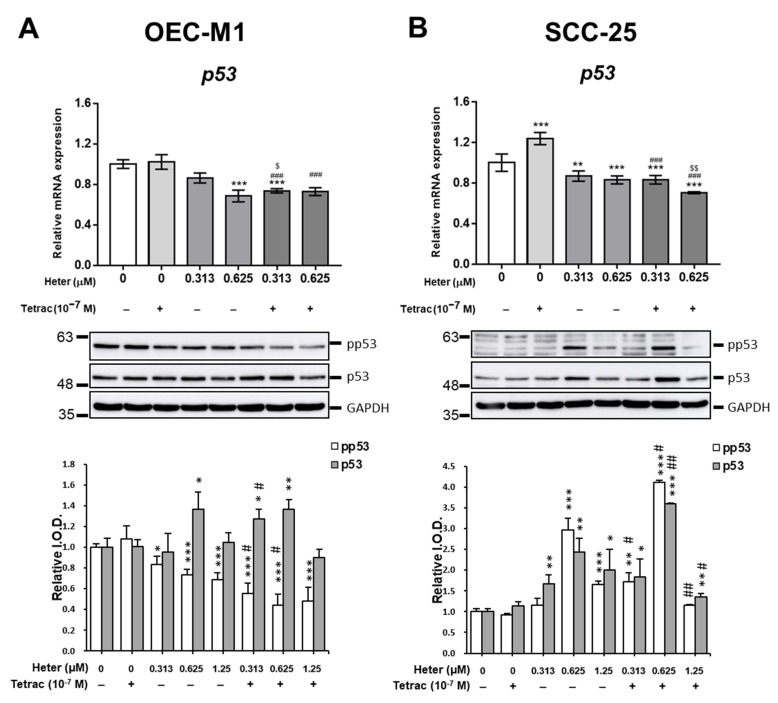
Tetrac and heteronemin regulate *p53* expression, p53 accumulation, and phosphorylation in oral cancer cells. OEC-M1 (**A**) and SCC-25 (**B**) cells were treated with 10^−7^ M tetrac, different concentrations of heteronemin (0.313 µM to 1.25 µM), or their combination for 24 h. Cells were harvested and RNA was extracted. qPCR was conducted for *p53* (upper panel). (*n*) = 4. Western blot analyses were conducted for phpspho-p53 and p53. GAPDH was used as an internal control (lower panel). Number of independent studies (*n*) = 4. * *p* < 0.05, ** *p* < 0.01, *** *p* < 0.001 compared to the control; ^#^
*p* < 0.05, ^##^
*p* < 0.01, ^###^
*p* < 0.001 compared to the tetrac treatment only; ^$^
*p* < 0.05, ^$$^
*p* < 0.01 compared to the same dosage of heteronemin treatment only.

**Figure 7 marinedrugs-18-00348-f007:**
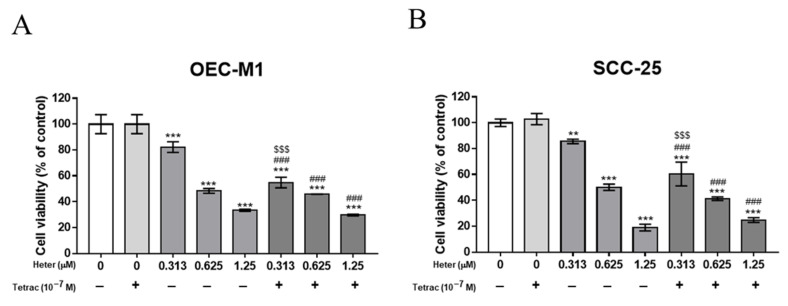
Combined treatment with heteronemin and tetrac suppressed proliferation in oral cancer cells. OEC-M1 (**A**) and SCC-25 (**B**) cells were treated with tetrac and different concentrations of heteronemin for 48 h. Cell proliferation was detected with a Cell Counting Kit-8. Number of independent studies (*n*) = 3. ** *p* < 0.01, *** *p* < 0.001, compared to the control; ^###^
*p* < 0.001, compared to the tetrac treatment only; ^$$$^
*p* < 0.001 compared to the same dosage of heteronemin treatment only.

**Figure 8 marinedrugs-18-00348-f008:**
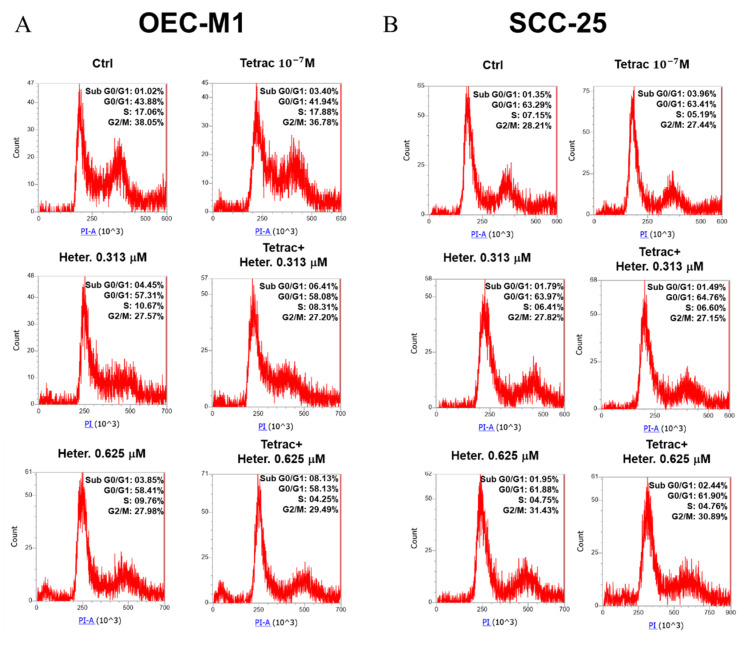
Combined treatment with heteronemin and tetrac induced cell cycle arrest in oral cancer cells. OEC-M1 (**A**) and SCC-25 (**B**) cells were treated with different concentrations of heteronemin in the presence or absence of 10^−7^ M tetrac for 24 h. Cells were harvested, and flow cytometric assay was conducted as described in the Materials and Methods section. (*n*) = 3.

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
