# Peer review of "Combined Treatment of Heteronemin and Tetrac Induces Antiproliferation in Oral Cancer Cells"

_marinedrugs, 2020, doi:10.3390/md18070348_

Round 1

Reviewer 1 Report

see attached file

Author Response

Reviewer 1.

In this manuscript, the authors investigated the anti-proliferative effects of the combined treatment of tetrac (thyroid hormone deaminated analogue) and heteronemin (a marine sesterterpenoid-type natural product), in oral cancer cells and further explored their molecular targets and partial mechanisms. Oral cancer is indeed a fatal disease, and its incidence is increasing in some parts of the world. Moreover, this cancer type is not the most studied cancer and any new suggested treatment and its mode of action is important. However, there some major comments to this study that should be addressed:

  1. The whole results section is written in the way describing the results as increase/decrease in the effect of each of the tested compounds alone or in combination; However, this is a scientific manuscript and should give accurate values in “X fold increase/decrease”; It will also be much easier to follow the results and its meaning without the need to calculate these values alone. Therefore, the authors should rewrite the whole manuscript; giving real numbers to the effects of both compounds.

Thanks for the reviewer's suggestion. We agree that giving accurate values in the results is easier for reader to figure out the effect of both compounds in this study. However, giving all numbers of the compounds in results section might make the article too miscellaneous to read. We did add accurate values in fold changes for increase and decrease. The Figures with numbers were in appendixes in the revised submission. 

  1. Along with the whole manuscript; there is a real need to "connect" paragraphs mainly in the results sections, to understand why the next step was performed. Sometime one additional sentence can make a huge difference in the understanding; Just one example: line 134-9: what actually should the combined treatment compensate for???

Thanks for the reviewer's comment. We totally agree with it. We apologize for not writing the statement clear enough. The paragraph line 134-9 was rewritten as follows. “Additionally, thyroid hormone stimulates TGF-β expression in oral cancer cells [15] which can be inhibited by tetrac. Thus, the combined treatment of tetrac and heteronemin might compensate for heteronemin-induced anti-proliferative activities via suppressing TGF-β and enhancing p53 expression in oral cancer cells.”. We also rewrote other unclear statements in the resubmitted version.

  1. In this regard; I would suggest reconsidering the name of the manuscript, for a more straightforward name. Talking about combination treatment of the two compounds for oral cancer.

Thanks for the reviewer's suggestion. The revised manuscript was retitled “The combination treatment of heteronemin and tetrac in oral cancer cells”.

  1. The authors should add control assays with both compounds, alone and in combination, to demonstrate that at the concentrations examined; heteronemin and tetrac have no effect on normal cells (fibroblasts or lymphocytes); such cell lines can be purchased from any cell culture organization.

Thanks for the reviewer’s suggestion. Tetrac doesn’t affect cell proliferation has been shown in the earlier studies that tetrac does not affect proliferation in CV-1 and L929 cells (PLoS Comput Biol. 2011 Feb 3;7(2):e1001073.).  Studies by Chong’s group revealed that heteronemin significantly suppressed the growth of human lung A549 cells, glioblastoma U87 cell; however, normal gingival fibroblast (HGF) cells were not affected (Respirology (2018) 23 (Suppl. 2), 90–334, AP383). In addition, HA nanoparticles/HET aggregates showed much weaker viability-inhibitory effects on L929 normal fibroblasts (Int J Nanomedicine. 2016 Mar 29;11:1237-51.).

  1. In figures 1&8 and in the result section describing these two experiments; there is no mention as to the sub-G0/G1 cell population; seen very clearly in Figure 1 at the high concentration of Heter. This cell population is a direct indication for apoptotic cells; should be evaluated and added and of course can add to the discussion regarding the death mechanism these two compounds are inducing.

Thanks for the reviewer’s suggestion. We analyzed the results of Figure 1 & 8 to demonstrate the population of sub-G0/G1 after both compounds treatment. Those results were presented in the revised manuscript.

  1. The discussion section is not added any valuable information behind the results; but rather is repeating the results; the authors should rewritten this section; talking about general aspects; such as advantages and disadvantages of this suggested treatment as compared to a specific targeted therapy known today for oral cancer.

Thanks for the reviewer's suggestion. We did discuss the role of TGF-β and p53 in cancer in our first submission. In the revised submission, part of discussion was re-written to discuss the potential of combined treatment of heteronemin and tetrac in oral cancer.

  1. Names of figures: please recheck and re-name the figures to match the real experiment; see for example Figure 1: “Heteronemin induces antiproliferation and gene profiles in oral cancer cells”; no gene profiles are tested and demonstrated here!

Thanks for the reviewer's suggestion. Title of “Heteronemin induces antiproliferation and gene profiles in oral cancer cells” was replaced as “Heteronemin induces antiproliferation and cell cycle arrest in oral cancer cells”.

  1. Figure 8-the same.

Thanks for the reviewer's suggestion. The Figure was changed.

  1. Abstract: Should be more general and highlighting the real outcome of the research; mainly in the results and conclusion (take part of the conclusions at the end of the manuscript) parts; Methods part are not giving real methods; the whole abstract should be rewritten.

Thanks for the reviewer's comments. The abstract was rewritten to match the reviewer's suggestion in the revised submission.

  1. Introduction: the first paragraph (lines 47-55); is for lay people and not scientists; either the authors should give some real information such as which kind of chemotherapy is mainly used today for oral cancer treatment; including references from the last years, which kind of “targeted therapy” is being tested for oral cancer, again with references, and ect; otherwise this paragraph should be cut out.

Thanks for reviewer’s comments. The paragraph was removed in the revised manuscript.

  1. Check English and coherence of sentences; for one example (there are many..): “It accounts for the fourth-highest incideno medical centers are stage III or IV cancer lesions”….???

Thanks for the reviewer's comments. Typos and grammar errors were corrected in the revised manuscript. Actually, there was a large piece of paragraph missing compared to our original manuscript.

It accounts for the fourth-highest incidence of malignancy in males and the seventh highest in the general population of Taiwan [2]. About 95% of oral cancer in Taiwan is oral squamous cell carcinoma (OSCC). Regrettably, about 50% of new OSCC cases presenting to medical centers are stage III or IV cancer lesions.

Shown on the website is as follows: It accounts for the fourth-highest incideno medical centers are stage III or IV cancer lesions.

Reviewer 2 Report

The studies conducted on the effects of heteronemin and tetrac in oral cancer cells was well carried out.

Some statements in the manuscript can be modified for better clarity.

For example, this statement on lines 47 to 48 is not clear: “It accounts for the fourth-highest incideno medical centers are stage III or IV cancer lesions.”

Lines 160 - 161 can be rephrased: “but it did not much help for phosphorylation status in combined treatment.”

Paragraph on lines 185-191 can be improved for better presentation

Problem with figure 8 caption on lines 234-238 – A,B,C – where is “C”?

Minor Typos – check the manuscript for typos, for example:

Line 89: “anti-proliferative” not ant-proliferative

Line 271: “natural products” not “nature products”

Line 282: “proliferation” not “roliferation”

Line 356: “Flow Cytometer” not “Flow Cytomter”

Author Response

Reviewer 2

The studies conducted on the effects of heteronemin and tetrac in oral cancer cells was well carried out.

  1. Some statements in the manuscript can be modified for better clarity. For example, this statement on lines 47 to 48 is not clear: “It accounts for the fourth-highest incideno medical centers are stage III or IV cancer lesions.”

Thanks for the reviewer's comments. Typos and grammar errors were corrected in the revised manuscript. Actually, there was a large piece of paragraph missing compared to our original manuscript.

It accounts for the fourth-highest incidence of malignancy in males and the seventh highest in the general population of Taiwan [2]. About 95% of oral cancer in Taiwan is oral squamous cell carcinoma (OSCC). Regrettably, about 50% of new OSCC cases presenting to medical centers are stage III or IV cancer lesions.

Shown on the website is as follows: It accounts for the fourth-highest incideno medical centers are stage III or IV cancer lesions.

We corrected it in the resubmitted paper.

  1. Lines 160 - 161 can be rephrased: “but it did not much help for phosphorylation status in combined treatment.”

Thanks for the reviewer's comments. The phrase was rewritten as follows.  Furthermore, heteronemin combined with tetrac showed the same trend to inhibit ERK1/2 activation. However, there was no synthetic effect since the inhibitory effect was maximal. These results indicated that both heteronemin and tetrac inhibited ERK1/2 activation and heteronemin further blocked STAT3 phosphorylation. The inhibitory effects by heteronemin and tetrac may play important roles in anti-proliferation in oral cancer cells.

  1. Paragraph on lines 185-191 can be improved for better presentation

Paragraph was rewritten as follows.

Heteronemin did not affect TGF-β1 expression significantly, but tetrac treatment and the combination treatment inhibited the expression of TGF-β1 in OEC-M1 cells (Figure 5).  Both tetrac treatment and combination treatment inhibited the expression of genes involved in angiogenesis and metastasis, THBS-1, and MMP9. On the other hand, heteronemin alone didn't affect the expression of THBS-1 and MMP9 in SCC-25 cells. Heteronemin increased expression of TGF-β1 in SCC-25 cells but this inductive effect of heteronemin was compensated in the combination with tetrac treatment (Figure 5).

  1. Problem with figure 8 caption on lines 234-238 – A,B,C – where is “C”?

C was a misplace which was removed in the revised submission.

  1. Minor Typos – check the manuscript for typos, for example:

Line 89: “anti-proliferative” not ant-proliferative

anti-proliferative replaced the wrong one.

Line 271: “natural products” not “nature products”

natural products replaced the wrong one.

Line 282: “proliferation” not “roliferation”

We apologize for the unnecessary typo. It was corrected

proliferation

Line 356: “Flow Cytometer” not “Flow Cytomter”

It is correct spelling as trademark The Attune™ NxT Flow Cytometer

Round 2

Reviewer 1 Report

The authors respond to all comments and corrected the manuscript accordingly.